# Prevalence of Cigarette Smoking and Influence of Associated Factors among Students of the University of Banja Luka: A Cross-Sectional Study

**DOI:** 10.3390/medicina58040502

**Published:** 2022-03-31

**Authors:** Ivana Todorović, Feng Cheng, Stela Stojisavljević, Sonja Marinković, Stefan Kremenović, Pane Savić, Ana Golić-Jelić, Nataša Stojaković, Svjetlana Stoisavljević-Šatara, Rajko Igić, Ranko Škrbić

**Affiliations:** 1Vanke School of Public Health, Tsinghua University, Beijing 100084, China; todorovici@who.int (I.T.); fcheng@tsinghua.edu.cn (F.C.); 2Association of Medical Students, Faculty of Medicine, University of Banja Luka, 78000 Banja Luka, The Republic of Srpska, Bosnia and Herzegovina; sonja.marinkovic@med.unibl.org (S.M.); stefankremenovic94@gmail.com (S.K.); panesavic@gmail.com (P.S.); 3Public Health Institute of The Republic of Srpska, 78000 Banja Luka, The Republic of Srpska, Bosnia and Herzegovina; stela.stojisavljevic@med.unibl.org; 4Center for Biomedical Research, Faculty of Medicine, University of Banja Luka, 78000 Banja Luka, The Republic of Srpska, Bosnia and Herzegovina; 5Department of Pharmacy, Faculty of Medicine, University of Banja Luka, 78000 Banja Luka, The Republic of Srpska, Bosnia and Herzegovina; ana.golic@med.unibl.org; 6Department of Pharmacology, Toxicology and Clinical Pharmacology, Faculty of Medicine, University of Banja Luka, 78000 Banja Luka, The Republic of Srpska, Bosnia and Herzegovina; natasa.stojakovic@med.unibl.org (N.S.); svjetlana.stoisavljevic-satara@med.unibl.org (S.S.-Š.); 7The Academy of Sciences and Arts of The Republic of Srpska, 78000 Banja Luka, The Republic of Srpska, Bosnia and Herzegovina; igicrajko@gmail.com

**Keywords:** cigarette smoking, students, prevalence, risk factors, secondhand smoke

## Abstract

*Background and Objectives*: Cigarette smoking among the youth population has increased significantly in developing countries, including Bosnia and Herzegovina. However, no extant literature assesses the prevalence of tobacco use, nor identifies factors associated with smoking. This study determined the prevalence of cigarette smoking among a specific cohort of students and assessed factors related to tobacco use in this population. *Materials and Methods*: This cross-sectional study included 1200 students at all faculties of Banja Luka University. Data were collected from questionnaires adapted from the Global Youth Tobacco Survey (GYTS) and the Global Health Professional Student Survey (GHPSS) standardized questionnaires and were analyzed using descriptive statistics, Pearson’s χ2 test, and logistic regression. *Results*: When the prevalence of cigarette smoking within the last thirty days was recorded, we found that 34.1% of students smoked within this period. Nearly three-quarters (74.9%) of the student population had smoked or experimented with cigarette smoking. However, medical students were 27.2% less likely to smoke than their counterparts from other faculties. Overall, 87% of all students were aware of the harmful effects of cigarette smoking, 79% were aware of the harmful effects of secondhand smoke, and 65% reported that it was difficult to quit. Increased spending of personal money was associated with a higher probability of smoking, while exposure to secondhand smoke increased the odds of smoking by 62%. *Conclusion*: Policies, strategies, and action plans should be introduced in order to reduce the prevalence of smoking among university students and to create a smoke-free environment at the various universities involved.

## 1. Introduction

Smoking is a major public health problem worldwide, resulting in many tobacco-associated deaths. It has been more than half a century since scientists first pointed out the dangers of tobacco use [1]. According to the World Health Organization (WHO), over one billion people are smokers globally, and 80% of them are from developing countries. It has been suggested that by 2030, tobacco smoking will kill more than eight million people each year [2]. Scientific data indicate the harmful effects of smoking and identify its contributions to the emergence of various types of cancers [3]. Smoking causes about 90% of all lung cancer deaths in the United States [4,5], and more women die from lung cancer each year than from breast cancer [6].

Trends in developed countries indicate that tobacco use decreases in adult populations, whereas in developing countries tobacco use increases as the population reaches adulthood [3]. According to data from the Republic of Srpska, the prevalence of smoking in a population aged from 15 to 65 was 24.6%, and tobacco use among the young population continues to rise [7]. The Global Youth Tobacco Survey (GYTS) results from 2013 and 2018 showed a significant increase over this time period in the use of tobacco and other tobacco products among 13 to 15 year old students, from 7.9% in 2013 to 9.1% in 2018. The percentage of students who had smoked at least once increased from 36.5% in 2013 to 38.9% by 2018 [7,8].

Worldwide, tobacco-use prevention measures remain focused on adolescents and young people. Although various social and behavioral factors of smoking have been identified globally, there is a need to understand the country-specific risk factors of smoking, especially among young people. The majority of adult smokers started smoking in their teenage or adolescent years, and it is essential to identify the factors associated with youth smoking behavior and take measures to reduce those [9]. In adolescence, cigarette smoking also contributes to other risk behaviors, such as risky sexual behavior, drug and alcohol consumption, and nutrition and dietary risks [9,10].

The aim of this study was to assess the prevalence of cigarette smoking among students at the University of Banja Luka, their attitudes concerning tobacco use and contributing factors associated with cigarette smoking.

## 2. Materials and Methods

This research was conducted as a cross-sectional survey at the University of Banja Luka, in the Republic of Srpska, Bosnia and Herzegovina, over April and May 2020. An adapted questionnaire, based on a standardized Global Health Professional Student Survey (GHPSS) and a Global Youth Tobacco Survey (GYTS), was used for data collection [11,12].

### 2.1. Sample Size

The young adult population of this study was estimated (CI of 95%) to establish that the sample size was no more than 5% at variance from the actual figure in the source population. Out of nearly 16,000 students from 16 faculties of the University of Banja Luka, a sample of 1200 students was used in our study.

### 2.2. Data Collection

Snowball sampling was used for data collection [13]. The first step of the survey was to identify and recruit at least one student representative from each faculty of the University of Banja Luka, to whom the research team had previously submitted an electronic questionnaire. These students were the initiators of the survey and disseminated the electronic questionnaire to others, the new participants in the survey. The questionnaire included the following: socio-demographic characteristics, use of tobacco products, exposure to secondhand smoke, smoking cessation, and attitudes regarding smoking.

### 2.3. Ethical Consideration and Confidentiality

This study was approved by the Ethics Committee of the University of Banja Luka (Decision No 18/4.17/19) and was conducted in accordance with the principles of the Helsinki Declaration [14]. Prior to entering this study, all participants were fully informed about the reasons for conducting the study, how their data would be used, were assured that anonymity would be maintained, and given assurance as to the risk-free nature of the study.

### 2.4. Statistical Analysis

The Statistical Package for the Social Sciences (SPSS), Windows version 25.0, was used to analyze the data obtained. Descriptive statistics analyzed demographic characteristics. The Pearson chi-square test was used for the frequencies, means, and the association of demographic variables with the prevalence of cigarette smoking. Multivariate logistic regression analysis was used to identify the relative importance of each predictor to the dependent variable by controlling the effects of other variables. The 95% confidence interval (CI) was estimated, and the *p*-value < 0.05 was considered significant.

## 3. Results

This study involved 1200 students aged between 18 and 26 years, with most students in the age group 20–21 (35.4%). The majority of study participants were medical students (41.4%) and the remainder were students from other faculties (58.6%). Most of the survey participants were females, almost twice the number of male students, which is in accordance with the gender distribution of students at the University of Banja Luka (Table 1).

A chi-square test of independence examined the relationship between demographic data and the prevalence of cigarette smoking among students. Of the five independent variables tested, only two were strongly associated with cigarette smoking; these were the faculty type and weekly expenses χ^2^ = 15.067, dF = 1, *p* < 0.001 and χ^2^ = 64.217, dF = 1, *p* < 0.001, respectively (Table 1).

Nearly three-quarters (74.9%) of the students we queried had tried cigarette smoking, while the remaining (25.1%) claimed that they had never smoked. The majority of smoking students (50.1%) were 16–18 years old when they first tried cigarettes, followed by those aged 11–15 years (25.3%). Although most individuals began smoking cigarettes before entering the university, a significant number (16.8%) experimented with cigarette smoking once they became students. We also evaluated the number of students who had smoked within the last thirty days. We noted that 34.1% of the group smoked within this period, while the remainder did not smoke at all (Table 2). Only 0.5% of smokers had used e-cigarettes or similar electronic devices. The results also showed that, out of the current smokers, most (69%) were ‘light smokers’, while the remainder were ‘moderate to heavy’ or ‘heavy’ smokers, and nearly two-thirds of students showed at least one sign of smoking addiction (Table 2).

Most subjects admitted that they had been exposed to cigarette smoking by others, and also that the majority of them were exposed to smoking at closed public places such as restaurants, cafes or bars (Table 3).

Analyses of questions related to students’ attitudes towards smoking and its relation to health revealed that most students acknowledged the harmfulness of cigarette smoking for health. As many as 77.4% of smokers and 93.3% of non-smokers believed that cigarette smoking is harmful to their health. We noted similar findings related to students’ concerns about the harmful effects of exposure to secondhand smoke. In fact, 66.1% of smokers and 85.6% of non-smokers were well aware that secondhand smoke is harmful to their health. The decision to use tobacco within the next twelve months generated an appreciable negative response, with at least half of participants (50.1%) claiming that they would not use tobacco compared to those who were unsure of their decision. When asked about the probability of using any tobacco products in the following year, 74.4 of non-smokers and 49.9% of smokers stated that they believed they will remain free of cigarettes (Table 4).

Various factors appear to be associated with smoking cessation among students. They avoided smoking because they wanted to improve their health (50%), to save money, and for other reasons that were personal to them. The majority of students (59.1%) admitted that they have never received advice regarding smoking cessation, while only 2.5% of students have been advised by health care professionals to quit smoking. The majority of students (88%) believed that they could quit smoking if they wished to, and more than 50% had tried to quit within the past 12 months. More than half of the smokers expressed an unwillingness to quit (Table 5).

We used a multivariable logistic regression to examine associated factors that could impact cigarette smoking. The model contained nine independent variables, and among them, six predictors were statistically significant in the single-factor logistic regression, and five independent variables were statistically significant in the multivariable logistic regression (Table 6). The multivariable logistic regression analysis showed that variables such as the study of medicine, available money, secondhand smoke at home, secondhand smoke in the faculty, and secondhand smoke in public spaces can determine whether students smoke or not. The variable ‘medical faculty students’ determined that medical students were 27.2% less likely to smoke than students from other faculties. The variable ‘more money available’ determined that if the students had sufficient income, the possibility of smoking increased by 12.4% (Table 6).

## 4. Discussion

The results of this study showed that the prevalence of cigarette smoking among the students within the last thirty days was 34.1%. The most important independent factors associated with cigarette smoking in the present study were the exposure to secondhand smoke, sufficient income, and the type of study students were engaged in. Additionally, this survey revealed that 74.9% of the participants have tried or experimented with cigarette smoking at least once, which is similar to the results of studies from Lithuania, Poland, and Slovakia but higher than in Belarus and Russia [15]. Our own data relate to students’ attitudes and beliefs regarding tobacco and tobacco product use and indicates that most (87.9%) of our cohort admitted that cigarette smoking is very harmful to their health, similar to the Polish study [16].

Whatever the leading risk factor is, the prevalence is higher than in the general population in the Republic of Srpska (24.6%) and among school-age children (9.1%) [7,8]. However, it should be noted that the above-mentioned studies involved people of a wider range, including adults from 15 to 65 years, and adolescents from 13 to 15 years, while our study was limited to university students aged from 18 to 26 years. The present study also suggested a link between the age of onset of smoking and current smoking status. Specifically, there was a higher prevalence of smoking among students who were aged 16–18 years old when they first tried cigarettes, followed by those aged 11–15 years, as well as a significant number of students experimenting for the first time with cigarette smoking at the age when they entered the university. These results are consistent with other studies conducted in Yemen [17], Turkey [18], Brazil [19], Ethiopia [20], Greece [21], and Serbia [22] in which an association of age with cigarette smoking was noticed. The strongest association with cigarette smoking in our study was the exposure to secondhand smoke at home, faculties, and closed public places, which increase the possibility of smoking by 62.3%. Our results are consistent with previous studies that showed a significantly higher occurrence of smoking amongst students exposed to secondhand smoke [23,24,25].

Another interesting finding of our study clearly showed that students with sufficient money are more likely to smoke cigarettes, increasing the possibility of smoking by 12.4%. This is similar to the results of the study in Yemen, a low-income country where family income is a significant predictor of smoking among university students [17]. Additionally, the study conducted in Turkey found a direct relationship between cigarette smoking and socioeconomic status [26]. In most studies, teens from families of higher socio-economic status were at additional risk for cigarette, alcohol, and substance use than those from households of lower socio-economic status [17,26]. In contrast, in developed countries such as Germany and Hungary, it was found that one’s financial situation had no association with any tobacco consumption among students [27].

We found that the smoking prevalence among medical students was statistically lower (27.2%) than those from other faculties (38.5%), similar to results among university students in Serbia [23] and Greece [15,28]. A systematic international review of tobacco smoking habits among medical students showed that low smoking rates were found in Australia and the United States of America (USA), while Spain and Turkey reported higher rates [29]. Medical students in Italy had a higher prevalence (37%) than students in the USA (6%) [30]. In China, a remarkable variance was found between the rates of smoking by the general population (66%) and medical students (3–6%) [31].

This study showed a higher smoking prevalence with almost equal proportions among both genders. In contrast, the studies from Yemen [17], Cameroon [32], Saudi Arabia [33], Ethiopia [34], and Greece [21] found that most tobacco consumers were males, who are especially influenced by their peers. Similarly, in China, tobacco use by boys (16.5%) was significantly higher than by girls (1.9%) [35]. A Georgian investigation showed a high smoking prevalence, with significant gender differences—male smokers making up 65% and females 35% [36]. Female medical students may have a lower smoking rate due to various factors such as religion, cultural and ethnic specificity, or socio-demographic differences, for example, in Asia or Africa where smoking is considered socially unacceptable for women, or in some Muslim countries, such as Bahrain or Saudi Arabia, where it is considered as an insult to custom [29]. On the contrary, the results of the GYTS study from Pakistan found that females were more likely than males to be susceptible to smoking [37]. The finding that female students are increasingly prone to cigarettes is important to understanding the evolving gender role associated with smoking initiation among adolescents. The WHO reported that the prevalence of smoking among females is increasing and women are a major target of opportunity for the tobacco industry [38].

Although many studies investigated the prevalence and risk factors associated with cigarette smoking such as age, gender, or peer influence, only a few studies addressed more than one risk factor simultaneously in their research, the exceptions included studies from Gambia [39], Bangladesh [40], Cameroon [32], Saudi Arabia [33] and Ethiopia [34]. Some of the main predictors of cigarette smoking in those studies included peer influence, male gender, age, parental influence (smoking allowed at home), awareness/attitudes/beliefs (poor knowledge of the harmful effects of smoking), intellectual pressure, sufficient pocket money, superiority complex, mass advertisement, attending grant-aided schools, and religion (non-Muslims).

It is interesting to note that most of our students had never been advised about smoking cessation. Despite several anti-smoking activities regularly performed by the medical students at our University in collaboration with the Public health Institute of the Republic of Srpska, such as the National Day Without a Cigarette (January 31st), World No-Tobacco Day (May 31st), and other measures, the rate of smoking remains high, and smoking is socially acceptable in this European region [41]. Since some non-medical and medical students start smoking during their first year at the university, this is the time when anti-smoking activity should be maximized. This is in line with previous worldwide findings that medical students smoke more in the later than in the earlier years of study [42]. This indicates that a higher prevalence of knowledge does not necessarily reduce the rate of smoking. Despite all anti-smoking actions and policies in developing countries, smoking prevalence remains high, with electronic cigarettes contributing additional numbers [22]. Because teenagers, in general, ignore any kind of anti-smoking information, a more effective approach might include strategies to support smoke-free environments.

In addition, tobacco control programs and prevention strategies should be oriented toward the student population [43]. From a public health perspective, students should become familiar with prevention measures. In order to reduce the number of smokers, university health authorities and medical students should develop a specific plan, including the creation of a Stop Smoking Center (Cessation Center) within the university health care system. This study should be used as a tool to support national health authorities to adopt laws banning tobacco smoking in public places. In Bosnia and Herzegovina, pursuant to Article 3 of the Law, it has been clearly stated that the legislator imposed a general prohibition of tobacco smoking in public spaces, including educational institutions, any institution used by children, school students, and university students, health care institutions and care providers, social institutions, and any other public institution.

Our study has several limitations. First of all, this cross-sectional study on students is not representative of students of the University of Banja Luka since the majority of interviewed participants were medical students. The second limitation of this study is that it relies on the self-reporting of smoking behavior, which is subject to recall bias. Thirdly, the study has focused only on tobacco smoking; no detailed information was collected regarding non-smoking tobacco use or illicit drug use, and these behaviors need to be explored in the future.

## 5. Conclusions

Our study showed that there is a very high prevalence of cigarette smoking amongst students at the University of Banja Luka, with almost equal proportions among both genders. Smoking prevalence among medical students was significantly lower than those from other faculties. The prevalence of cigarette smoking appears to be closely related to the availability of money, as well as to the particular faculty (medical vs. non-medical students), and exposure to secondhand smoking. A large majority of the cohort admitted that cigarette smoking is very harmful to their health, but most of our students had never been advised about smoking cessation. These findings emphasize the importance of implementing policies, strategies and action plans aimed at supporting a smoke-free environment.

## Figures and Tables

**Table 1 medicina-58-00502-t001:** Students’ smoking status and socio-demographic characteristics.

Variables	Smoking			
	Yes (%)	No (%)	χ^2^	*p*-Value *
Gender				
Male	34.7	65.3	0.081	0.777
Female	32.8	67.2		
Age				
≤19 years	27.3	72.7	5.843	0.558
20 years	34.6	65.4		
21 years	35.9	64.1		
22 years	34.2	65.8		
23 years	32.8	67.2		
24 years	38.1	61.9		
25 years	28	72		
≥26 years	38.6	61.4		
Year of faculty				
year 1	29	71	8.037	0.090
year 2	38.5	61.5		
year 3	37.3	62.7		
year 4	31.3	68.7		
year ≥ 5	31.6	68.4		
Faculties				
Medical students	27.8	72.2	15.067	<0.001
Non-medical students	38.5	61.5		
Weekly Expenses				
Do not have	22.1	77.9	64.217	<0.001
<10 BAM **	29.3	70.7		
11–20 BAM	41.5	58.5		
21–30 BAM	49.4	50.6		
31–40 BAM	47.8	52.2		
41 BAM and more	24.2	75.8		

* *p*-value significant at ≤ 0.05; χ^2^ = chi-square. ** BAM is the abbreviation of the official currency of Bosnia and Herzegovina (1 Euro = 1.95 BAM).

**Table 2 medicina-58-00502-t002:** Students’ cigarette smoking characteristics.

Ever experimented with Cigarette Smoking		%
Yes		74.9
No		25.1
Age at first experience with cigarette smoking		%
<10		7.8
11 to 15		25.3
16 to 18		50.1
>19		16.8
Current cigarette smokers *	*N*	%
Yes	410	34.1
No	790	65.9
Smoking status related to the number of cigarettes/day		%
Light smokers (1–10 cigarettes/day)		69.0
Moderate-to-heavy smokers (11–20 cigarettes/day)		24.1
Very heavy smokers (>20 cigarettes/day)		6.9
Has a desire to smoke after waking up		%
Always		9.5
Sometimes		25.1
Never		65.4
Smoking time after waking up		%
Within 60 min		20.9
1 to 2 h		16
2 to 4 h		9.8
More than 4 h, less than one day		9.3
1 to 3 days		5.1
4 days and more		4.2
Never		34.7
Signs of addiction **		%
Yes		61.4
No		38.6

* Students who smoked at least one day in 30 days prior to the survey. ** Students who smoke every day and desired to smoke always or sometimes upon waking up, and where smoking time after waking up was within one day.

**Table 3 medicina-58-00502-t003:** Students’ exposure to secondhand smoke.

Exposure to Secondhand Smoke	No %	Yes %
Exposed to secondhand smoke (total)	21.3	78.7
Exposed at home	52.2	47.8
Exposed at university	44.0	56.0
Exposed at closed public places	5.5	94.5

**Table 4 medicina-58-00502-t004:** Students’ attitudes concerning tobacco use and exposure to secondhand smoking.

Questions	Probability	Non-Smokers	Smokers	Total	χ^2^	df	*p*-Value
		%	%	%			
Do you think tobacco smoking is harmful to your health?	Definitely not	0.6	1.2	0.8	65,962	3	0.000
	Probably not	0.1	1.7	0.7			
	Probably yes	6.0	19.6	10.6			
	Definitely yes	93.3	77.4	87.9			
Do you think exposure to secondhand smoke is harmful to your health?	Definitely not	0.2	1.2	0.5	72,818	3	0.000
	Probably not	0.1	3.5	1.3			
	Probably yes	14.1	29.2	19.2			
	Definitely yes	85.6	66.1	79.0			
Once someone has started smoking tobacco, do you think it will be difficult for them to quit?	Definitely not	2.0	2.7	2.3	14,266	3	0.003
	Probably not	8.5	14.9	10.7			
	Probably yes	65.9	64.0	65.2			
	Definitely yes	23.6	18.4	21.8			
Are you in favour of banning smoking at faculties and other education facilities?	Yes	94.9	75.5	88.4	98,045	1	0.000
	No	5.1	24.5	11.6			
Are you in favour of banning smoking at restaurants?	Yes	92.3	48.5	77.4	368,823	1	0.000
	No	7.7	51.5	22.6			
Are you in favour of banning smoking at cafes and bars?	Yes	80.2	51.5	60.8	292,892	1	0.000
	No	19.8	100.0	39.2			
Are you in favour of banning the advertising of tobacco and tobacco products?	Yes	78.6	62.5	73.1	34,836	1	0.000
	No	21.4	37.5	26.9			
Do you think you will use any tobacco product in the next 12 months?	Definitely not	74.4	2.2	50.0	873,565	1	0.000
	Probably not	21.2	9.4	17.2			
	Probably yes	2.8	44.8	17.0			
	Definitely yes	1.6	43.6	15.8			

**Table 5 medicina-58-00502-t005:** Students’ attitudes related to smoking cessation (*n* = 410).

Questions	Yes %	No %		
Would you like to stop smoking?	43.6	56.4		
Did you try to stop smoking in the last year?	51	49		
Do you believe you will be able to stop smoking when you decide?	88	12		
The reason why someone would like to stop smoking	Health %	Finances %	Friends do not like it %	Family does not like it %
	50.0	18.8	0.5	30.7
Did you receive any help in smoking cessation?	Yes, from a health professional (%)	Yes, from a friend or family member (%)	Someone else (%)	No (%)
	2.5	33.1	5.3	59.1

**Table 6 medicina-58-00502-t006:** Logistic regression analyses.

Variables	Univariable		Multivariable	
	*p*-Value	OR (95% CI)	*p*-Value	OR (95% CI)
Gender	0.777	0.964 (0.746–1.245)		
Age	0.362	1.08 (0.915–1.276)		
Year of faculty	0.995	1.00 (0.910–1.099)		
Medical students	<0.001	0.613 (0.478–0.785)	0.023	0.728 (0.554–0.957)
More money available	<0.001	1.340 (1.201–1.494)	0.004	1.193 (1.058–1.346)
Secondhand smoke at home	<0.001	1.385 (1.284–1.494)	<0.001	1.195 (1.099–1.300)
Secondhand smoke at faculty	<0.001	1.575 (1.429–1.736)	0.004	1.182 (1.056–1.323)
Secondhand smoke in public spaces	<0.001	1.946 (1.745–2.169)	<0.001	1.623 (1.435–1.837)
Banning smoking in buildings	0.007	1.409 (1.096–1.811)		

## Data Availability

All data presented in this study are available upon request.

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
