# Peer review of "Prevalence of Cigarette Smoking and Influence of Associated Factors among Students of the University of Banja Luka: A Cross-Sectional Study"

_medicina, 2022, doi:10.3390/medicina58040502_

Round 1

Reviewer 1 Report

This is a well-designed and executed survey of university cigarette smoking in a country with above-average poverty.  Results were consistent with previous studies of similar target populations.  Recommendations did not go beyond the data and human subjects protections were adequate.  

Weaknesses:  Two weaknesses stand out.  First is the omission of any items referring to experience/use of eCigarette/vaping.  Thus it is unclear if students may be nicotine addicting by alternative means to tobacco smoking.  Second it a lack of analysis of second-hand smoke exposure.  Specifically, whether exposure increases risk of smoking.  Correlations do not confirm causality.  It is likely that students who smoke actively seek roommates who smoke, and students who do not smoke seek roommates who do not smoke.  Thus, exposure alone offers little information about the role of second-hand smoke and "contagion." 

Finally, I found it disturbing that medical school faculty tolerate student smoking at all.  This is contrary to accepted public health policy internationally.  Healthcare providers who smoke provide a bad example to their patients - giving the impression that smoking is not dangerous.  The link between smoking and reduced life expectancy has been shown to be causal.  This result calls into question the university's curriculum addressing the very many diseases associated with tobacco smoking - most of which reduce life expectancy.  This is the opposite of health promotion one expects from all healthcare providers - especially physicians. 

Reviewer 2 Report

  1. The study aim should be more precise "The aim of this study was...."
  2. Study sample - please provide more precise data on population (faculties at the university and its share in the total sample of recruited subjects)
  3. Please attach an English version of the questionnaire as supplementary material or precisely describe questions that were used to assess the smoking status
  4. The results section is too extensive - please avoid describing all the results in the text. Please provide the most important results in the text, and the rest of them are present in Tables.
  5. In table 5 please provide thr number of subjects (n=.....)
  6. Tables 6 and 7 are unclear and should be more precise (Especially confidence intervals)
  7. Please provide practical implications of this study
  8. Please provide a conclusion based on obtained findings and please avoid overwhelming conclusions
  9.  

Round 2

Reviewer 2 Report

Thank you for addressing all the comments. The manuscript was professionaly revised.
